# MCP-YOLO: A Pruned Edge-Aware Detection Framework for Real-Time Insulator Defect Inspection via UAV

**DOI:** 10.3390/s25227049

**Published:** 2025-11-18

**Authors:** Hongbin Sun, Shijun Guo, Xin Pan, Qiuchen Shen, Yaqi Xu, Jianchuan Ma, Zhanpeng Qu

**Affiliations:** 1School of Electrical and Information Engineering, Changchun Institute of Technology, Changchun 130012, China; 20241041223@stu.ccit.edu.cn (S.G.); 20231041209@stu.ccit.edu.cn (Q.S.); xuyaqi@stu.ccit.edu.cn (Y.X.); majianchuan@stu.ccit.edu.cn (J.M.); 20241041230@stu.ccit.edu.cn (Z.Q.); 2School of Computer Technology and Engineering, Changchun Institute of Technology, Changchun 130012, China; panxin@ccit.edu.cn

**Keywords:** deep learning, defect detection, YOLO11, transmission lines, smart grid

## Abstract

Unmanned Aerial Vehicle (UAV)-based inspection of transmission line insulators faces significant challenges due to complex backgrounds, variable imaging conditions, and diverse defect characteristics. Existing deep learning approaches often fail to balance detection accuracy with computational efficiency for edge deployment. This paper presents MCP-YOLO (Multi-scale Complex-background Pruned YOLO), a lightweight yet accurate detection framework specifically designed for real-time insulator defect identification. The proposed framework introduces three key innovations: (1) MS-EdgeNet module that enhances multi-granularity edge features through grouped convolution, improving detection robustness in cluttered environments; (2) Dynamic Feature Pyramid Network (DyFPN) that combines dynamic upsampling with re-parameterized multi-branch architecture, enabling effective multi-scale defect detection; (3) Auxiliary detection head that provides additional supervision during training while maintaining inference efficiency. Furthermore, Group SLIM pruning is employed to achieve model compression without sacrificing accuracy. Extensive experiments on a real-world dataset of 3091 UAV-captured images demonstrate that MCP-YOLO achieves 92.1% mAP@0.5, 90.5% precision, and 89.0% recall, while maintaining only 8.65 M parameters. Compared to state-of-the-art detectors, the proposed method achieves superior detection performance with significantly reduced computational overhead, reaching 250 FPS inference speed. The model size reduction of 37.3% from the baseline, coupled with enhanced detection capabilities, validates MCP-YOLO’s suitability for practical deployment in automated power grid inspection systems.

## 1. Introduction

Transmission lines constitute critical infrastructure within modern power systems, facilitating long-distance transmission of high-voltage electricity [1]. Insulators serve as core components of transmission lines, providing both electrical insulation and mechanical support. Their performance stability directly impacts0 grid insulation strength and power transmission efficiency, playing a pivotal role in suppressing corona discharge and maintaining electromagnetic environments, particularly in ultra-high-voltage lines of 500 kV and above. However, prolonged exposure to adverse weather conditions and intense electromagnetic fields renders insulators susceptible to various defects, including mechanical breakage, electrical flashover, material aging, and pollution-induced surface degradation [2]. These issues not only diminish operational efficiency [3,4] but also pose significant risks potentially leading to large-scale blackouts and substantial economic losses. Therefore, timely and accurate detection of insulator defects remains crucial for maintaining grid stability and minimizing economic impact [5].

Existing object detection methods can be categorized into traditional manual feature extraction and deep learning-based approaches [6]. Manual detection, despite widespread adoption, exhibits notable limitations including low efficiency, high labor costs, and susceptibility to human error. With rapid advances in artificial intelligence and machine vision, deep learning-based object detection algorithms have demonstrated significant application potential through superior image analysis performance. Recent emergence of drone-based inspection systems has revolutionized transmission line maintenance. However, detecting defects in drone-captured images presents considerable challenges due to complex backgrounds, dense objects, small target sizes, and occlusions [7,8]. The quality of UAV images—prerequisite for reliable defect detection—is highly dependent on the acquisition process. Recent studies have emphasized that optimizing the UAV’s 3D flight trajectory and controlling camera perspectives can minimize inspection time for large-scale infrastructure detection while ensuring consistent image resolution [9,10]. While advances in these acquisition strategies are crucial, the core challenge addressed in this study lies in the algorithmic detection of defects in images already captured from complex environments. The proposed model is designed to remain robust to persistent challenges (such as complex backgrounds, small targets, and occlusions) that still exist even in well-acquired images. Additionally, drone deployment platforms face resource constraints, and as detection accuracy improves, model sizes increase correspondingly. Balancing detection accuracy with model size represents another critical challenge requiring attention. Therefore, optimizing and adapting state-of-the-art algorithms to meet specific requirements of transmission line defect detection remains an important research focus with substantial practical significance.

To address these challenges, we propose the Multi-scale Complex-background detection and Pruning (MCP) mechanism. A fast and accurate network specifically designed for UAV-based object detection. Built upon the YOLOv11 framework with targeted improvements, our approach achieves both high precision and lightweight performance, enabling more accurate and efficient object detection in UAV systems. This model serves as a multi-task enhanced detector for industrial inspection, with its core innovation residing in balancing accuracy and speed for small objects and occluded scenarios.

The primary contributions of this paper are as follows:In the backbone network, we introduce MS-EdgeNet to replace residual blocks in C3K2. The Multi-Scale Edge Information Enhancement Module (MS-EdgeNet) fundamentally strengthens the network’s perception of multi-granularity edge features through combining edge enhancement at multiple scales. This improves the model’s adaptability to complex scenes, enabling superior detection of targets and occluded objects in challenging environments. Additionally, grouped convolution reduces parameter count while preserving spatial design integrity.The DyFPN module enhances the Neck structure by combining the DySample module [11] with the RepGFPN module from DAMO-YOLO [12]. RepGFPN employs a multi-branch structure during training through re-parameterization to strengthen feature fusion capabilities. By integrating DySample, information flows more flexibly across different levels. This improves the model’s detection performance for objects of varying sizes, particularly in scenarios involving occlusion or small targets.An auxiliary detection head (Auxiliary Head) [13] is incorporated at the detection head position. The original YOLOv11 lacks an auxiliary head, resulting in absence of additional supervision during training. The auxiliary detection head enhances gradient flow during training, particularly for multi-scale objects or occlusion cases. During training, the auxiliary detection head improves the model’s generalization capability and detection accuracy, while during inference, the auxiliary head is removed to reduce computational load and memory usage [14].To achieve model lightweighting, the Group SLIM pruning method [15,16] is employed for model compression. This approach reduces model size without requiring modifications to the original detection architecture. It not only enhances the model’s generalization capability and robustness by mitigating overfitting risks but also effectively addresses hardware resource constraints and computational efficiency imbalances faced by UAV platforms in small object detection within complex scenarios, achieving dual optimization of algorithm performance and deployment environment.

In experiments, the MCP-YOLO model demonstrates efficient and rapid detection of insulator conditions on transmission lines in complex scenarios and under occlusion, providing excellent real-time performance and robustness. This design achieves high detection accuracy in industrial inspection while ensuring real-time inference speeds suitable for practical applications.

## 2. Related Work

With rapid advances in artificial intelligence and machine vision, deep learning-based object detection algorithms have demonstrated significant application potential through superior image analysis performance. Deep learning automatically extracts data features through multi-layer neural networks and finds widespread application in image recognition. Deep learning-based object detection models primarily fall into two categories. The first category encompasses models such as RetinaNet [17], the SSD series [18], EfficientDet [19], DETR [20], and the YOLO series [21,22,23,24,25]; the second category comprises two-stage models including R-CNN [26], Fast R-CNN [27], Faster R-CNN [28] and Mask R-CNN [29].Two-stage models achieve high detection accuracy through precise localization, yet their complex structure and heavy computational load result in slow inference speeds, rendering them unsuitable for deployment on resource-constrained drones. Conversely, single-stage models directly regress bounding boxes, offering faster inference with fewer parameters, thus proving more suitable for deployment on resource-limited drones. Among these, YOLO represents a groundbreaking algorithm in object detection. Its core concept transforms the detection task into a single regression problem, directly predicting target bounding boxes and class probabilities across entire images. By implementing end-to-end processing with a single neural network, YOLO significantly improves computational speed, making it particularly suitable for real-time applications. YOLOv1 [21] pioneered single-stage detector concepts, capable of predicting bounding boxes and class probabilities directly from input images in a single forward pass, revolutionizing object detection. YOLOv3 [22] achieves multi-scale prediction through adopting a feature pyramid structure, enhancing detection capability for small objects, while the C3 module in YOLOv5 improves efficiency while maintaining strong feature representation [30]. Additionally, YOLOv8 introduces a more efficient C2F module and decoupled head structure, rendering the model more lightweight while preserving effective feature representation, proving particularly advantageous for drone deployment. YOLO11 represents the latest iteration in the YOLO series of object detection models. It explores architectural innovations, including introduction of C3k2 (cross-stage with kernel size) blocks, SPPF (spatial pyramid pooling fast), and C2PSA (convolutional block with parallel spatial attention) components, thereby improving feature extraction in multiple dimensions and contributing to enhanced model performance.

Several studies have explored YOLO model applications for insulator detection in UAV deployments. Wang et al. proposed a specialized plugin adapted for YOLO algorithms, enhancing insulator breakage detection accuracy from YOLOv4 to YOLOv8 [31]. Qu et al. successfully reduced model parameters by designing lightweight feature pyramid networks and lightweight head networks, enabling deployment on edge devices for real-time insulator defect detection [32]. He et al. introduced MFI-YOLO [33], which improved multi-type insulator fault detection accuracy in complex backgrounds through lightweight multi-scale feature extraction (MSA-GhostBlock). Lu et al. proposed IDD-YOLO, integrating a lightweight attention mechanism called LCSA (Lightweight Channel-Spatial Attention) with GhostNet to capture features more comprehensively, achieving an actual inference speed of 20.83 frames per second (FPS) [34]. To enhance defective insulator detection efficiency, Luo et al. [35] designed an adaptive anchor box extraction method. Gao et al. [36] proposed a cloud-edge collaborative intelligent strategy for insulator identification and defect detection scenarios. Li et al. [37] introduced lightweight EGC convolution, significantly reducing model complexity through optimizing parameter structures. Sun et al. [38] proposed ID-Det, a novel insulator defect detection model, and introduced the Insulator Clipping Module (ICM). Huang et al. [39] proposed an algorithm that integrates knowledge distillation to streamline the insulator detection model.

A common challenge throughout these works involves difficulty balancing model accuracy and parameter size in complex scenarios, often leading to overfitting risks. To address these issues, this paper proposes MCP-YOLO, aiming to achieve high-precision deployment with even more limited resources while reducing overfitting risks and improving real-time performance.

## 3. Method

### 3.1. MCP-YOLO Structural Framework

To address the unique challenges of detecting insulator defects in transmission lines from drone-captured images, we propose MCP-YOLO. The architecture of MCP-YOLO is illustrated in Figure 1. To effectively tackle the aforementioned challenges, MCP-YOLO is designed with the following key capabilities:(1)Multi-scale edge enhancement capability to address the challenge of detecting small and occluded targets in complex backgrounds—This is achieved through our C3k2-MS module, which replaces the Bottle Neck in C3K2 with MS-EdgeNet, fundamentally strengthening the network’s perception of multi-granularity edge features by integrating edge details across different scales.(2)Dynamic multi-scale feature fusion capability to handle varying sizes of insulator defects and partial occlusions—This capability is realized by our DyFPN module, which combines DySample’s lightweight dynamic upsampling with RepGFPN’s multi-branch structure, enabling flexible information flow across different levels.(3)Enhanced training supervision capability to improve generalization under small-sample conditions and prevent overfitting—This is implemented through our Auxiliary Head, which provides additional supervision signals during training to enhance gradient flow, while being removed during inference to maintain efficiency.(4)Lightweight deployment capability to ensure real-time performance on resource-constrained UAV platforms—This is enabled by Group SLIM pruning method, which achieves multi-level compression while maintaining detection accuracy.

As shown in Figure 1. Building upon the YOLO11 framework, MCP-YOLO integrates these capabilities through systematic architectural improvements: the C3k2-MS module enhances the backbone’s feature extraction, the DyFPN module replaces the original neck for superior multi-scale fusion, and the Auxiliary Head augments training supervision. Following model optimization through Group SLIM pruning, which enables layer skipping and structural compression, the final MCP-YOLO model achieves significant reductions in parameters (from 13.79 M to 8.65 M), memory usage, and inference latency. Through this integrated design, MCP-YOLO achieves the critical balance of maintaining high-precision insulator defect detection (mAP@0.5: 0.921) while enabling real-time inference (250 FPS) on resource-constrained UAV platforms, effectively solving the fundamental challenge of deploying sophisticated detection algorithms in practical power line inspection scenarios.

### 3.2. Detailed Introduction to the C3k2-MS Module

To address challenges of detecting small and occluded targets, this paper introduces the MS-EdgeNet module to replace residual blocks in C3K2. Building upon C3K2′s original functionality, it enhances multi-scale edge information by specifically replacing Bottle Neck with the MS-EdgeNet module. Unlike original YOLOv11, module selection depends on the C3k parameter—choosing either C3-MS-EdgeNet or standard MS-EdgeNet modules. When C3k is True, the multi-scale edge enhancement module with C3k structure is utilized; otherwise, the standard MS-EdgeNet module is employed. The number of sub-modules is controlled by parameter n, enabling depth-wise stacking. The EdgeNet module (structure shown in Figure 2) enhances edge information in input features through residual learning. Its core concept highlights edge features via high-frequency residual learning. First, the input feature map x undergoes average pooling to obtain edge information, where blurring preserves primarily low-frequency components. A differential operation then retains high-frequency information, followed by processing edge features through a convolutional layer with Sigmoid activation function. Finally, the original input is combined with processed edge features to generate the final output.

The MS-EdgeNet module (structure shown in Figure 3) enhances model performance in complex scenes through multi-scale feature extraction and edge information enhancement, integrating edge details across different scales. Its core design comprises multi-scale branches for capturing contextual information at varying granularities, edge enhancement for strengthening local edge features in each branch, and final feature fusion to consolidate multi-scale information for optimized output.

First, input the feature map x ∈ R^(B×C×H×W)^, where the number of channels is C = inc. A 3 × 3 convolution (Local Conv) is used to directly process the original features, with the formula as follows:(1)$$\mathrm{LocalFeat}(x)={\mathrm{Conv}}_{3\times 3}(x)\;\in R^{B\times C\times H\times W},$$

Extract local details of the original features through 3 × 3 convolution and store them as baseline features in the list out. Then proceed with multi-scale branch processing. For each branch i ∈ {1, 2, …, n}, reduce the spatial dimensions to bin [i] using adaptive average pooling (AdaptiveAvgPool2d), with the formula as follows:(2)$$x_{\mathrm{pool}}^{\left(i\right)}={\mathrm{AdaptiveAvgPool}}_{b_{i}}(x)\;\in R^{B\times C\times b_{i}\times b_{i}},$$

Downsample the input to the target size bi × bi to capture the global context at scale bi. A 1 × 1 convolution reduces the number of channels to inc/len (bins), with the formula being:(3)$$x_{\mathrm{down}}^{\left(i\right)}={\mathrm{Conv}}_{1\times 1}(x_{\mathrm{pool}}^{\left(i\right)})\;\in R^{B\times (C/N)\times b_{i}\times b_{i}},$$

At the same time, a 3 × 3 depthwise separable convolution is used to further process the features, with the number of parameters being 3 × 3 × (C/N).(4)$$x_{\mathrm{dw}}^{\left(i\right)}={\mathrm{DepthwiseConv}}_{3\times 3}(x_{\mathrm{down}}^{\left(i\right)})\;\in R^{B\times (C/N)\times b_{i}\times b_{i}},$$

Then, the feature map is restored to the original size H × W through bilinear interpolation upsampling. The formula is:(5)$$x_{\mathrm{up}}^{\left(i\right)}=\mathrm{Interp}(x_{\mathrm{dw}}^{\left(i\right)},H,W)\;\in R^{B\times (C/N)\times H\times W},$$

Finally, this branch enhances edge information through the EdgeEnhancer module.(6)$$x_{\mathrm{edge}}^{\left(i\right)}=\mathrm{EdgeEnhancer}(x_{\mathrm{up}}^{\left(i\right)})\;\in R^{B\times (C/N)\times H\times W},$$

Then save the results.(7)$$\mathrm{out}=\left[\mathrm{LocalFeat}(x),x_{\mathrm{edge}}^{\left(1\right)},\ldots ,x_{\mathrm{edge}}^{\left(N\right)}\right],$$

Concatenate all branch outputs along the channel dimension, merging all branches:(8)$$x_{\mathrm{concat}}=\mathrm{Concat}\left(\mathrm{LocalFeat}(x),x_{\mathrm{edge}}^{\left(1\right)},\ldots ,x_{\mathrm{edge}}^{\left(N\right)}\right)\;\in R^{B\times (C+N\cdot (C/N))\times H\times W}=R^{B\times 2C\times H\times W},$$

Integrate information through 1 × 1 convolution (Final Conv):(9)$$y={\mathrm{Conv}}_{3\times 3}(x_{\mathrm{concat}})\;\in R^{B\times C\times H\times W},$$

The 3 × 3 convolution compresses the number of channels from 2C back to C, achieving multi-scale feature fusion.

### 3.3. Detailed Introduction to the DyFPN Module

In the initial YOLOv11 model, the UpSample method demands substantial computational resources and numerous parameters [40]. To address issues of inaccurate boundary localization and poor detection performance for small objects, this paper adopts DySample, a lightweight dynamic upsampling approach achieving efficient upsampling through point sampling rather than traditional dynamic convolution. Traditional upsampling blurs small object features, leading to missed detections or misclassifications. DySample dynamically shifts to capture local details, enhancing small object feature representation. It supports two distinct upsampling modes (lp and pl). In lp (Low-Process) mode, the offset is learned first. If scope is enabled, sampling range is controlled via scope (x). sigmoid, then combined with the offset and finally added to the initial position before invoking the sample method to execute sampling operations. This ensures low memory consumption while maintaining computational complexity linear to input resolution, suitable for real-time applications. However, offset generation remains constrained by low-resolution information, limiting detail preservation. In pl (Post-Process) mode, pixel shuffle is first applied to rearrange input x. The offset is then computed, and convolution results are restored to original scale via pixel unshuffle, with offset adjusted based on scope enablement. Finally, the sample method performs sampling. The advantage lies in capturing high-frequency details and achieving fine-grained control, particularly suitable for handling sharp edges and textures. However, it requires storing high-resolution intermediate features, resulting in high memory consumption. Since computational complexity increases quadratically with output resolution, this leads to significant computational demand increases. In the DySample module, the sampling operation is the core component, which utilizes learned offsets to dynamically sample the input feature map. This process is implemented through the sample method. The offsets are learned through convolutional layers, representing per-pixel displacement values. In the sample method, the offsets are first reshaped into the form (B, 2, −1, H, W), where B is the batch size, 2 denotes horizontal and vertical offsets, and H and W are the height and width of the input feature map, respectively. A coordinate grid is generated, where ‘coords_h’ and ‘coords_w’ represent vertical and horizontal coordinates, ranging from [0, H−1] and [0, W−1], respectively. Adding 0.5 ensures the coordinates correspond to the center of the feature map. A 2D grid of coordinates is created, representing the (x, y) positions of each point in the feature map. The dimensions are adjusted to meet the requirements of subsequent operations. Finally, the coordinates are converted to the input data type, moved to the GPU, and normalized. [W, H] denotes the width and height of the input feature map. The formula is:(10)$$coord_{norm}=2\times \frac{\left(coord_{phy}+\Delta \right)}{\left(W,H\right)}-1,$$

Perform pixel rearrangement on the coordinates. This is the key operation for upsampling, where pixel rearrangement is used to adjust the coordinates to fit the upsampled size, enabling the final sampling operation. The input feature map x and the adjusted coordinates coords are paired, and bilinear interpolation (mode = ‘bilinear’) is applied to sample the input feature map. By combining pixel rearrangement and grid sampling, the dynamic upsampling operation based on learned offsets is ultimately achieved. The core advantage of using DySample lies in its ability to adaptively adjust sampling positions according to the input feature map and the learned offsets, thereby improving the quality and flexibility of upsampling.

The CSPStage in Figure 1 is a crucial component of RepGFPN. It serves as the concrete implementation form of CSPNet’s cross-stage partial network. Its core objective is to reduce computational redundancy through channel splitting and residual connections [41], while simultaneously improving feature reuse efficiency. The specific structure is illustrated in Figure 4.

The input feature map x is split into two parts via two 1 × 1 convolutions: a direct path (with channel number C_first = C_out/2) and a residual path (output channel number ch_mid = ch_out − ch_first). This can be mathematically expressed as: The input feature map x is split into two parts via two 1 × 1 convolutions: a direct path (with channel number C_first = C_out/2) and a residual path (output channel number ch_mid = ch_out − ch_first). This can be mathematically expressed as:(11)$$\left\{ \begin{array}{l} y_{1}={\mathrm{Conv}}_{1\times 1}(x;W_{1})\in R^{H\times W\times C_{\mathrm{first}}}\;,\\ y_{2}^{\left(0\right)}={\mathrm{Conv}}_{1\times 1}(x;W_{2})\in R^{H\times W\times C_{\mathrm{mid}}}\;, \end{array}\right.$$

The residual pathway passes through n BasicBlock_3 × 3_Reverse blocks, each containing a 3 × 3 depthwise separable convolution to reduce computational cost. The residual connection enables gradient propagation across layers, mitigating gradient vanishing. If spp = True, a Spatial Pyramid Pooling (SPP) module is inserted at the intermediate position (after the (n−1)/2-th block) to enhance the receptive field through multi-scale pooling. Mathematically, this can be expressed as: for each block k = 1, 2, …, n.(12)$$y_{2}^{\left(k\right)}=\left\{\begin{array}{l} {\mathrm{Block}}_{k}\left(y_{2}^{\left(k-1\right)}\right),\\ \mathrm{SPP}\left({\mathrm{Block}}_{k}\left(y_{2}^{\left(k-1\right)}\right)\right), \end{array}\right.$$

When there is no SPP or k is not an intermediate block, y^2(k)^ selects the former; when k is an intermediate block and SPP is enabled, y^2(k)^ selects the latter. Finally, concatenate y1 with all intermediate results:(13)$$\mathrm{concat}\;\mathrm{feat}=\left[\begin{array}{c} y_{1}+y_{2}^{\left(1\right)}+y_{2}^{\left(2\right)}+\cdots +y_{2}^{\left(n\right)} \end{array}\right],$$

Finally, the channels are adjusted through a 1 × 1 convolution and the result y is output.

Our model’s Neck section, compared to the original YOLO11 Baseline’s Neck, has been upgraded from 4 C3k2 modules to 5 CSPStage modules. Although the increased number of modules leads to a rise in overall computational load, the CSPStage design is more efficient—its optimized channel splitting reduces redundant calculations. This means performance improves without a significant increase in total computation. Additionally, the five modules enhance the model’s depth, enabling it to capture more complex features and strengthen multi-scale feature integration, making it more effective for small object detection.

We propose the DyFPN module, which combines DySample (a dynamic upsampling module) with CSPStage multi-stage joint architecture (alternating stacking of CSPStage and DySample) to form an efficient and flexible feature extraction-upsampling joint framework, demonstrating significant advantages in dense prediction tasks. Compared to the original bilinear interpolation upsampling [42], our DySample employs dynamic kernel generation and transforms feature fusion from single-path concatenation to multi-path intermediate feature concatenation. While this increases computational complexity, it substantially enhances feature representation capability. This design significantly improves multi-scale feature interaction through dense cross-layer connections and dynamic upsampling, particularly excelling in detecting small and occluded objects in complex scenes. Although computational costs rise, real-time performance can still be maintained through structural optimizations (such as channel splitting in CSPStage) and hardware acceleration.

### 3.4. Introduction to Detect-Aux

The Detect-Aux in Figure 1 (detailed structure shown in Figure 5) represents an improved detection head for YOLOv11, which incorporates an auxiliary detection head (Auxiliary Head) alongside the standard detection head to enhance model training through multi-branch supervision. It dynamically generates anchors by automatically adjusting anchor points and stride based on input feature map resolution. Primary Head outputs target bounding boxes (Box) and class probabilities (Class). Auxiliary Head participates in computation only during training phase, providing additional supervision signals. The auxiliary head maintains identical structure to the primary head but operates with independent parameters and processes features from different Neck layers. Outputs from both primary and auxiliary heads contribute to loss calculation, offering dual supervision. Gradients from the auxiliary head backpropagate through the shared Backbone network, enhancing feature representation. This dual gradient backpropagation mitigates vanishing/exploding gradient issues, stabilizing the training process. The auxiliary head is subsequently removed during inference to reduce computational overhead. Dynamic anchor generation supports inputs of varying resolutions, improving multi-scale adaptability. Below is the mathematical formula for the anchor generation function (make anchors):(14)$$\left\{\begin{array}{l} grid_{x}=\mathrm{arange}(0,W)\cdot \mathrm{stride},\\ grid_{y}=\mathrm{arange}(0,H)\cdot \mathrm{stride},\\ \mathrm{anchors}=\mathrm{stack}(({\mathrm{grid}}_{x},{\mathrm{grid}}_{y}),\dim=-1), \end{array}\right.$$

Among them, H and W are the height and width of the feature map.

By further integrating multi-scale features through a feature pyramid, optimized feature maps 1, 2, and 3 (all part of a single image) are generated. During the training phase, both the primary head and auxiliary head are activated: the outputs of both participate in loss calculation. During the inference phase, the auxiliary detection head is removed. The Primary Head is a multi-branch parallel structure composed of multiple branches from Cv2 and Cv3, with each sub-branch processing input features from different levels, though each branch independently handles a single input feature map. Cv2 primarily functions for bounding box regression, predicting the positional offsets (center coordinates (cx, cy) and dimensions (w, h)) of target bounding boxes. Through DFL (Distribution Focal Loss), the continuous bounding box offsets are modeled as discrete probability distributions to improve regression accuracy. The input comes from multi-scale feature maps in the Neck, and the output predicts 4 * reg_max values for each anchor, representing the discrete distribution of the four parameters of the bounding box. Assuming reg_max = 20, the final offset is calculated via weighted summation, which can be mathematically expressed as:(15)$$\mathrm{offset}=\overset{19}{\underset{i=0}{\sum }}P_{i}\cdot i,$$

The primary role of cv3 is to perform category classification, predicting the probability distribution of the target belonging to each class. It maintains computational efficiency through depthwise separable convolution and outputs multi-label classification results via the Sigmoid activation function. The input consists of the same feature maps as cv2, while the output predicts nc values (where nc is the number of classes) for each anchor point, representing class probabilities. The classification output is activated by the Sigmoid function, which can be mathematically expressed as:(16)$${\mathrm{cls}}_{p}\mathrm{rob}=\sigma ({\mathrm{cls}}_{l}\mathrm{ogits})\in {\left[0,1\right]}^{C},$$
where C is the number of categories.

The advantage of this design is that during task decoupling, regression and classification are separated to prevent mutual interference between the two tasks. During optimization, they are independently optimized through different loss functions (DFL + BCE) for the regression and classification branches, respectively. Each input feature map has independent Cv2 and Cv3 branches, adapting to object detection at different resolutions. The outputs of both branches are concatenated to form the complete detection results, achieving end-to-end multi-object detection.

The primary function of the auxiliary detection head is to generate auxiliary detection results through an additional prediction branch, which participates in loss calculation alongside the main detection head. This enhances the model’s ability to detect multi-scale objects and challenging cases such as small or occluded objects. Unlike the main detection head, which processes the first n1 layers of features output by the Neck, the auxiliary detection head processes the last n1 layers of features from the Neck. The auxiliary detection head Cv4 handles the regression branch, while Cv5 handles the classification branch. The specific mathematical formulation can be expressed as:(17)$${\mathrm{aux}}_{b}\mathrm{ox}={\mathrm{dfl}}_{a}\mathrm{ux}(\mathrm{cv}4(x)),$$

Return to branch Cv4, generate bounding box distribution parameters, and decode them via dfl aux.(18)aux cls=σ(cv5(x)),

The classification branch Cv5 generates category probabilities through Sigmoid activation. The design of the auxiliary detection head can enhance small object detection, provide intermediate supervision signals, and accelerate model convergence. Meanwhile, the main head and auxiliary head supervise features at different levels, respectively, forcing the network to maintain high discriminability across multiple scales. The auxiliary head is only activated during training and completely removed during deployment, thus not affecting inference speed. The total loss is the weighted sum of the main head loss and auxiliary head loss, which can be mathematically expressed as:(19)$$L_{\mathrm{total}}=L_{\mathrm{primary}}+\lambda \cdot L_{\mathrm{auxiliary}},$$
where λ is the auxiliary loss weight (typically λ ∊ [0.2, 0.5])

In experiments, these improvements enable the model to better detect small and occluded objects in complex scenes, making it highly effective for insulator detection in power transmission lines. The next section details the experimental setup and evaluation metrics used to validate the effectiveness of the proposed enhancements.

### 3.5. Lightweight Adjustment of MCP-YOLO

When deploying object detection models on drones, embedded devices must perform model inference. Generally, factors affecting practical model deployment include model size, runtime memory usage, and computational requirements. During object detection model inference, the activation response process of convolutional neural networks consumes significant runtime memory, imposing substantial resource burden on embedded devices. This paper employs the hybrid sparse pruning method Group SLIM [15,16] to sparsify and lightweight the trained model, utilizing a simple yet effective network retraining scheme to achieve deployment on resource-limited drone edge devices. The Group SLIM pruning method combines Group Lasso [15] and Network Slimming [16]. During sparsification, Group Lasso penalizes L2 norm on a group-wise basis to achieve group-level sparsity. which can be mathematically expressed as:(20)$$\underset{\beta }{\min}\left(∥y-\overset{L}{\underset{l=1}{\sum }}X_{l}\beta_{l}{∥}_{2}^{2}+\lambda \overset{L}{\underset{l=1}{\sum }}\sqrt{p_{l}}∥\beta_{l}{∥}_{2}\right),$$

Here, y represents the dependent variable vector, Xℓ denotes the coefficient vector of the ℓ-th group, and L indicates the total number of groups. λ stands for the regularization strength parameter (controlling the degree of sparsity), and pℓ is the weight of the ℓ-th group.

Network Slimming achieves channel-level sparsity through L1 regularization of the scaling factors in the BN layer. Its formula can be expressed as:(21)$$L=\underset{\left(x,y\right)}{\sum }l(f(x,W),y)+\lambda \overset{K}{\underset{k=1}{\sum }}|\gamma_{k}|,$$

Here, γk is the BN layer scaling factor for the k-th channel, |γ_k_| is the L1 norm of the scaling factor, K is the total number of channels in the network, and λ controls the channel sparsity strength.

The two methods are combined: first, redundant groups are eliminated through Group Lasso, and then Network Slimming is used to prune channels within the groups, achieving efficient multi-level compression. Additionally, removing non-critical channels may cause a temporary decline in model performance. Therefore, if pruning is applied to the model, fine-tuning should be employed to restore performance fluctuations, allowing the model to regain its previous inference capabilities.

This experiment employs layer-wise pruning, retaining important layer channels to prevent excessive pruning. The speed-up parameter is set to 1.55, indicating 55% improvement in post-pruning inference speed, with sparse regularization strength coefficient reg set to 0.0005. The model undergoes 500 rounds of sparse training before pruning to ensure sufficiently prolonged training stabilizes channel importance ranking and avoids random pruning. After completing the pruning process, the new network demonstrates more compact structure in terms of model size, memory consumption, and computation time while exhibiting superior performance compared to the initial detection model. This model lightweighting process proves typically repeatable—through multiple iterations, further multi-channel network slimming can be achieved, progressively refining its structure.

Through these proposed improvements, the model demonstrates enhanced capability for detecting small and occluded objects in complex scenes, proving highly effective for insulator detection in power transmission lines. The subsequent section details experimental setup and evaluation metrics used to validate the effectiveness of these enhancements.

## 4. Experiment

### 4.1. Datasets

The dataset employed in this study was collected through actual UAV inspection missions conducted on operational transmission lines in northern China, providing genuine real-world data rather than simulated scenarios. This authentic data collection approach significantly enhances the research value and practical applicability of our findings. The dataset encompasses various insulator types and defect conditions captured under diverse environmental challenges:(1)Background complexity: The images were captured across diverse terrains typical of northern China’s transmission line corridors, including dense forests with varying vegetation, agricultural farmlands with seasonal crop variations, mountainous regions with complex topography, urban-industrial areas with building interference, and open plains under different weather conditions. These varied backgrounds create significant detection challenges, particularly when insulators appear against cluttered or similarly colored backgrounds.(2)Insulator diversity: The dataset includes three main insulator types commonly used in China’s power grid: glass insulators with their characteristic transparent/green coloration, porcelain insulators featuring white/gray ceramic surfaces, and composite insulators with polymer housings. The color variations pose particular challenges—glass insulators often blend with vegetated backgrounds, white porcelain insulators can be indistinguishable against cloudy skies or snow, and weathered insulators exhibit discoloration that complicates defect identification.(3)Defect characteristics: The dataset captures two critical defect categories encountered in actual grid operations: flashover damage showing characteristic burn marks and surface degradation, and broken/cracked insulators with varying degrees of structural damage. These defects were identified and verified by experienced grid maintenance personnel during routine inspections.

This real-world data collection ensures that our model is trained on authentic operational challenges rather than idealized conditions, significantly improving its deployment readiness for practical power grid inspection applications.

The dataset comprises two defect categories: flashover damage and broken insulators. To ensure model generalization capability and acceptable performance, the dataset is partitioned as shown in Table 1.

This data partitioning strategy with a 7:2:1 ratio effectively mitigates overfitting while ensuring robustness to unseen data. Figure 6 displays representative insulator examples, with defective areas circled in red.

All experiments were conducted utilizing the PyTorch2.5.0 deep learning framework. The hardware configuration included an NVIDIA RTX 4060 GPU and a 12th Gen Intel Core i7-12700KF CPU. Key hyperparameters comprised an initial learning rate of 0.01, final learning rate of 0.1, batch size of 16 images per batch, and 200 training iterations. Data augmentation techniques, including mosaic augmentation with factor 1.0, were applied to enhance model generalization. To maintain consistency during training and evaluation, input images were resized to 640 × 640 pixels. The experimental configuration is detailed in Table 2.

### 4.2. Evaluation Metrics

In YOLO-based object detection tasks, evaluation metrics including mAP, precision, recall, F1 score, and model size prove crucial for assessing model performance.

Precision is a measure of the reliability of a model’s predictions. Its formula is:(22)$$Precision=\frac{TP}{TP+FP},$$
where TP and FP represent correctly and incorrectly detected objects, respectively.

Recall measures the model’s ability to identify actual targets, expressed as:(23)$$Recall=\frac{TP}{TP+FN},$$
where FN denotes ground truth objects that the model failed to detect.

Average Precision (AP) represents the area under the precision-recall (PR) curve for individual categories, reflecting model performance across different confidence thresholds. AP is calculated as the integral of the interpolated PR curve, expressed as:(24)$$AP=\int_{0}^{1}P\left(r\right)dr,$$

mAP is the average of AP scores across all object classes, which can be expressed as:(25)$$mAP=\frac{1}{N}\overset{N}{\underset{i=1}{\sum }}AP,$$
where N is the total number of classes.

The F1 score evaluates classification model performance, proving particularly useful when addressing imbalanced datasets. It combines precision and recall metrics through calculating their harmonic mean, with its mathematical expression being:(26)$$F1=2\times \frac{Precision\times Recall}{Precision+Recall},$$

The F1 score ranges from 0 to 1, with higher values indicating superior model performance. An F1 score of 1 indicates perfect precision and recall achievement, while 0 indicates extremely poor model performance.

Model size in deep learning refers to storage space occupied during storage or operation, directly affecting storage requirements on hard drives or in memory. Larger models demand increased computational resources and time for both training and inference. Its mathematical formula is:(27)$$\mathrm{Model}\;\mathrm{Size}\;(\mathrm{MB})=\frac{Params\times \;\mathrm{bytes}}{1024\times 1024},$$

Here, params represent the number of parameters, and bytes denote bytes. This experiment uses 4 bytes, meaning when calculating Model Size in this paper, the mathematical expression is:(28)$$\mathrm{Model}\;\mathrm{Size}\;(\mathrm{MB})=\frac{Params\times \;4}{1024\times 1024},$$

FPS (Frames Per Second) constitutes an important measure of model processing speed for video or image sequences in real time. It represents the number of frames processed per second, typically used to evaluate model performance in real-time applications; higher FPS values indicate faster image detection speeds.

### 4.3. Model Comparison Experiment

The model constructed in this paper represents a target detection model. To validate the performance of the proposed insulator defect detection algorithm, in addition to comparing with the baseline network, several state-of-the-art target detection models from recent years (YOLOv5, YOLOv6, YOLOv7, YOLOv8, YOLOv10, and RTDETR) were selected for experiments on the same dataset. Table 3 summarizes Precision, Recall, mAP@0.5, F1score, Model size and FPS for each algorithm. The normalized model bar stereogram and radar plot are presented in Figure 7.

Considering hardware limitations of our drone deployment platform, the required parameter count and computational load must be minimized. As shown in Table 2, MCP-YOLO significantly outperforms all comparative models with a frame rate of 250 FPS. Although the transformer-based RTDETR achieves higher Precision, Recall, and mAP@0.5 than other models, its massive parameter count and model size occupy substantial space, with Model size reaching 75.81 M, rendering it unsuitable for drone deployment. The proposed MCP-YOLO model achieves an optimal balance between detection performance and deployment efficiency. It attains the highest inference speed (250 FPS) and the second-highest mAP@0.5 (92.1%) with a minimal model size (8.65 M), demonstrating its practical superiority for UAV-based edge deployment. Compared to the larger YOLOv7, while Precision, Recall, and mAP@0.5 are only marginally higher, our model size represents merely 1/16th of YOLOv7—just 8.65 M—and operates over three times faster. When compared to the compact YOLOv10 (8.64 M), our model is only 0.01 M larger yet outperforms it by 7.7% in Precision, 8% in Recall, and 5.7% in mAP@0.5. Typically, smaller models exhibit lower performance; however, we overcome this trade-off. While maintaining extreme speed and lightweight design, MCP-YOLO still ranks among top performers in accuracy, achieving an mAP@0.5 of 92.1%, surpassing all variants except YOLOv7. Its high Precision (0.905) and Recall (0.89) underscore its robustness in minimizing false positives and demonstrate strong capability in effective object identification.

In experiments, our improved model demonstrates robustness and accuracy in insulator defect detection across various complex environments, including scenes with cluttered backgrounds, further validating its effectiveness and reliability in challenging scenarios. Thus, our improved model represents a significant advancement over standard YOLOv11. By integrating three advanced modules—C3 k2-MS, DYFPN, and Auxiliary Head—the model achieves high-accuracy detection in environments with visual noise, complex backgrounds, and for small targets. These enhancements enable the model to efficiently distinguish defects even when target-background contrast is low, ensuring reliable detection results under diverse conditions. Meanwhile, through Group SLIM pruning, the model not only enhances generalization ability and anti-interference by reducing overfitting risks but also effectively addresses adaptation challenges of limited hardware resources and computational efficiency imbalances faced by UAV platforms when detecting small targets in complex scenarios, achieving dual optimization of algorithm performance and deployment environment.

This study systematically evaluates model robustness and reproducibility in classification performance metrics through multiple rounds of co-training experiments, wherein optimized models undergo side-by-side comparison with existing detection models. Based on accuracy confidence interval analysis of cross-experiment statistics, the study quantifies detection efficacy of each model, thus providing statistically significant empirical support for improving model generalization ability. This multi-dimensional evaluation framework effectively reveals algorithm stability characteristics under data distribution variation, providing scientific basis for engineering deployment in industrial scenarios.

### 4.4. Ablation Experiments

To comprehensively evaluate performance gains of each optimization module on the model, ablation experiments were conducted on the proposed improved model using a dataset specifically designed to address unique challenges of detecting insulator defects in transmission lines. These experiments, based on the YOLOv11-n model, first examined each module’s contribution to the baseline by progressively integrating them individually, thereby clearly assessing specific impact of each module on detection accuracy and efficiency. The detailed experimental procedure follows: First, the C3k2 module in the network was replaced with the proposed C3k2-MS module; then, under identical conditions, the Neck section was replaced with the DyFPN module; subsequently, the original detection head was replaced with the dynamic detection head Detect-Aux while maintaining all other conditions unchanged; finally, sparse pruning training was incorporated. The contributions of these modifications to metrics including precision, recall, mAP, F1 score, model size, and FPS were evaluated. The specific ablation experiments are presented in Table 4.

First, in Experiment 1, the C3k2-MS module was introduced into the Backbone, resulting in 0.9% increase in mAP@0.5, 0.8% improvement in precision, and 2% boost in recall rate. While maintaining improvements in accuracy, recall rate, and mAP@0.5, Model Size decreased slightly from 9.85 M to 9.65 M, demonstrating C3k2-MS’s lightweight capability and preserved spatial design. The F1 score increased to 84.29, indicating the C3k2-MS module’s adaptability. This enables superior model detection of objects in complex scenes and occluded targets.

In Experiment 2, with all other conditions unchanged, the Baseline’s Neck was replaced with the DyFPN module. As previously mentioned, four C3k2 modules were replaced with five CSPStage modules. Model size increased from 9.85 to 13.99, yet recall rate improved by 4.2%, mAP@0.5 increased by 1.7%, and F1 Score reached 85.19%. This indicates that DySample enhances upsampled feature quality, while CSPStage strengthens feature extraction and fusion capabilities. Their combination enables the Neck section to preserve more useful information during feature transmission and fusion while suppressing noise, thereby improving overall detection performance.

In Experiment 3, with all other conditions unchanged, an auxiliary detection head (Auxiliary Head) was incorporated at the detection head position. Model Size remained unchanged, while Precision, Recall, and mAP@0.5 increased by 0.8%, 0.5%, and 1%, respectively. This indicates that the Auxiliary Head participates in training during the training phase but is removed during inference, reducing computational load and memory usage.

In Experiment 4, after introducing the C3k2-MS module into the Backbone and replacing the Neck with the DyFPN module, Precision increased by 2.3%, Recall by 4.2%, mAP@0.5 by 2.2%, and F1 Score reached 86.17%. Meanwhile, Model Size also decreased. These results demonstrate that combining C3k2-MS and DyFPN significantly enhances overall model performance, with their synergistic effect enabling superior feature extraction. This allows more effective model detection of targets and occluded objects in complex scenes.

In Experiment 7, based on introducing the C3k2-MS module into the Backbone, we replaced the Neck with the DyFPN module while incorporating an Auxiliary Head. Our model achieved excellent combined detection results, with Precision reaching 0.905—3.2% improvement over baseline—and Recall reaching 0.855, 6.7% higher than baseline. The mAP@0.5 reached 0.909, marking 4% increase over baseline. F1 Score achieved 87.93. However, Model Size increased to 13.79 M, and although FPS remained lower than baseline, due to strict drone deployment constraints, despite strong detection performance, it proves less conducive to practical deployment.

In Experiment 8, we introduced the Group SLIM pruning method, which removes redundancies, effectively “slimming down” and “purifying” the model. Compared to Experiment 7, Precision remained stable at 0.905, while Recall improved from 0.855 to 0.89. This indicates that the model’s feature extraction network became more focused on core, robust target features. Consequently, the model’s ability to recognize challenging positive samples (including small, occluded, or blurry targets) improved, detecting more previously missed True Positives (TP), significantly boosting Recall. Since Recall experienced substantial increase while Precision remained high, this almost inevitably led to expansion in PR curve area (i.e., AP/mAP). This constitutes the core driver behind the mAP@0.5 rise to 0.921.

In experiments, MCP-YOLO significantly reduces resource consumption (model size, computational load) while simultaneously improving core detection performance (Recall, mAP), with Precision maintained at high levels. Figure 8 normalization demonstrates different improved models’ performance regarding Precision, Recall, and mAP (still applying inverse normalization to Model size).

However, in Experiment 5, after introducing the C3k2-MS module into the Backbone and only adding the Auxiliary Head, both recall value and mAP@0.5 fell short of Baseline performance. Therefore, we investigated specific reasons behind this, ruling out overfitting. First, overfitting was excluded. Subsequent research revealed that the C3k2-MS module extracts edge detail features in shallow networks, emphasizing high-frequency information. Introducing auxiliary supervision (Auxiliary Head) in intermediate layers forces the network to learn high-level semantics before reaching deeper layers. This creates interference between high-frequency edge features in shallow layers and semantic features in intermediate layers, making it difficult for the network to balance details and semantics. With our DyFPN module introduction, structural reparameterization technology is employed. During training, multi-branch training with parallel cross-scale connections prevents shallow details from directly interfering with intermediate-layer supervision. During inference, it merges into a single path, retaining key features while suppressing noise. This enables dynamic fusion of shallow edge features and deep semantic features, assigning dynamic weights to feature maps of different scales. Consequently, the network focuses more on edges in shallow layers and semantics in intermediate layers, rather than forcing rigid mixing. In the Auxiliary Head’s intermediate supervision stage, input feature maps undergo DySample denoising, resulting in purer semantic information and reduced reliance on erroneous features. To improve issues in Experiment 5, we considered adding a channel attention mechanism (SE Block) before the Auxiliary Head to suppress irrelevant edge noise. The SE Block primarily employs channel attention mechanisms, dynamically learning each channel’s weight through global information to emphasize important channels and suppress redundant ones, thereby enabling the network to focus on more meaningful features.

The specific operation involves taking an input x with shape (B, C, H, W), compressing spatial information through Squeeze to obtain (B, C, 1, 1), and adjusting it to (B, C) for fully connected layer input. After passing through the fully connected layer, a weight vector y of shape (B, C) is obtained, which is then reshaped back to (B, C, 1, 1) for subsequent broadcast multiplication. Finally, the weight vector y is multiplied channel-wise with the original input x (x * y) to achieve channel feature recalibration. This process dynamically learns the weight of each channel through global information, emphasizing important channels and suppressing redundant ones, guiding the network to focus on more meaningful features. Table 5 presents results after introducing the SE Block [43] module. Experiment 10 demonstrates that although Precision decreases after adding the SE Block, Recall improves by 3% and 3.3% compared to Baseline and Experiment 5, respectively, validating our proposed improvement’s effectiveness. However, simply adding the SE Block module to Baseline may not prove optimal. YOLOv11 detection requires simultaneous attention to local details and global semantics. The SE module compresses spatial information through global average pooling, potentially over-smoothing local features and weakening spatial sensitivity required for detection tasks. Comparing Experiment 11 with Experiment 7, our final improved model with SE Block module shows 1.8% decrease in Precision but 0.9% increase in Recall. This indicates that SE Block amplifies important channel features (e.g., defect regions) while suppressing secondary channels (e.g., background). Specifically, it enhances edge and texture features related to defects (e.g., cracks, damage), making the model more sensitive to subtle and blurry defects, thereby reducing missed detections while also decreasing false positives caused by background interference. The Precision decrease may result from high-frequency edges or noise being misclassified as defects. Achieving balance between these factors and determining optimal method selection in industrial applications requires further research.

### 4.5. Visualization

The visual analysis in this paper employs heatmap visualization technology to further illustrate our improved model’s superiority in insulator defect detection. The highest activation values in heatmaps (red/yellow regions) should precisely cover insulator defect locations (including flashover points, defect areas, and metal connection points), aligning with YOLO detection box central regions. Metal connectors appear yellow because they constitute critical insulator structure components—flashovers frequently occur at metal-ceramic interfaces, prompting the model to identify them as defect-sensitive areas. Local red edges overlapping with YOLO-detected defect boxes indicate defect regions. The background appears deep blue without noise. Figure 9 presents a heatmap comparison between our improved model and baseline.

Comparative analysis across diverse insulator defect scenarios demonstrates the superior robustness and precision of MVP-YOLO. As shown in the first row of Figure 9, MVP-YOLO accurately detects flashover defects despite significant challenges, including color similarities between the flashover, insulator body, and background, as well as interfering contamination; it exhibits no confusion with contamination artifacts, highlighted by distinct markings for defects (red dashed) and uncontaminated regions (red solid). The second row of Figure 9 further reveals MVP-YOLO’s ability to precisely localize subtle, layered damage sites by focusing on regions exhibiting clear anomalous gradients, effectively suppressing background noise. While mainstream models detect flashover in the third row of Figure 9, MVP-YOLO uniquely mitigates spurious background activations prevalent in baseline models (evident as erroneous red dots), showcasing enhanced noise suppression. Under rigorous evaluation, MVP-YOLO consistently outperforms alternatives, establishing itself as a robust solution for defect detection in complex real-world environments.

## 5. Conclusions

This study proposes the MCP-YOLO model, utilizing drone-captured images specifically for defect detection in transmission line insulators. By integrating the C3k2-MS module, DyFPN module, and Auxiliary Head module, and optimizing into a sparse and lightweight structure through group slim, the model achieves comprehensive optimization from multi-scale feature analysis to precise target region localization. It effectively addresses challenges including small object detection, complex backgrounds and occlusions, and drone deployment constraints. Experimental results on a specially designed dataset for insulator defect detection demonstrate that the model achieves average precision of 0.905, recall of 0.89, and mAP@0.5 of 0.921, with model parameter size of only 8.65 M. Compared to existing models including YOLOv5, YOLOv6, YOLOv8, and YOLOv10, our model delivers superior performance, particularly in precision, recall, and mAP metrics. When benchmarked against relatively high-precision YOLOv7 and RTDETR models, our model surpasses YOLOv7 in key metrics while representing approximately 1/16 of its size. The RTDETR model is 8.77 times larger than ours, rendering our solution significantly more suitable for drone deployment.

The MS-EdgeNet module enhances model performance in complex scenes through multi-scale feature extraction and edge information strengthening, integrating edge details across different scales. The DyFPN module combines DySample with CSPStage in multi-stage joint architecture, forming an efficient and flexible feature extraction-upsampling hybrid framework demonstrating significant advantages in dense prediction tasks. The Auxiliary Head not only enhances detection capability but can be removed during inference, enabling the model to maintain high detection accuracy in complex scenarios while ensuring real-time inference speeds suitable for practical applications. These findings validate the proposed method’s effectiveness in improving transmission line defect detection reliability and efficiency. By employing Group SLIM pruning for model compression, the approach reduces overfitting risks, enhances generalization capability and anti-interference performance, while effectively addressing hardware resource constraints and computational efficiency imbalances faced by UAV platforms during small target detection in complex environments—achieving dual optimization of algorithm performance and deployment conditions.

Future research will focus on constructing generalized datasets encompassing multimodal defect samples and enhancing detection robustness under extreme weather conditions through developing environment-adaptive feature enhancement algorithms (including low-light denoising and rain-fog degradation modeling). Collaborative edge-cloud deployment with smart grid dynamic monitoring platforms will be implemented to achieve real-time diagnosis and early warning of insulator health status. Furthermore, validating the deployed performance of MCP-YOLO on dedicated edge computing platforms (e.g., NVIDIA Jetson modules) represents an immediate and critical step for practical application. Our future work will anchor in this direction, focusing on benchmarking and further optimizing the model’s efficiency on such resource-constrained hardware.

## Figures and Tables

**Figure 1 sensors-25-07049-f001:**
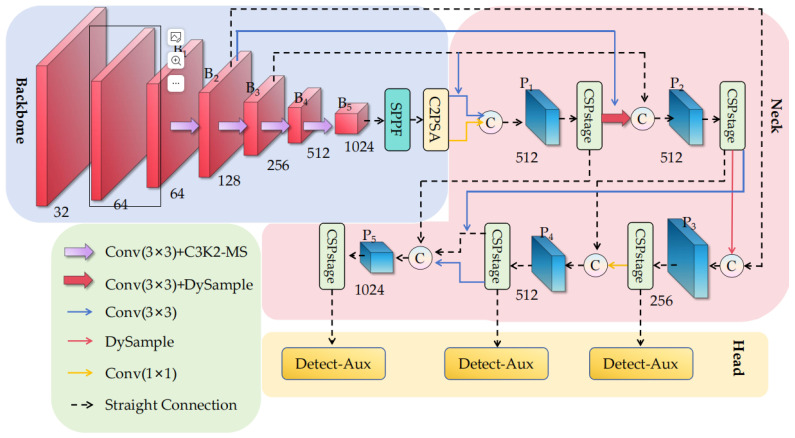
MCP-YOLO Architecture Diagram.

**Figure 2 sensors-25-07049-f002:**
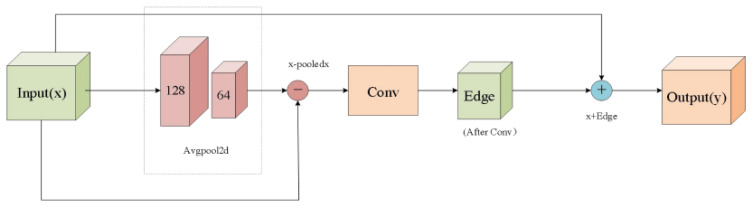
EdgeNet architecture diagram.

**Figure 3 sensors-25-07049-f003:**
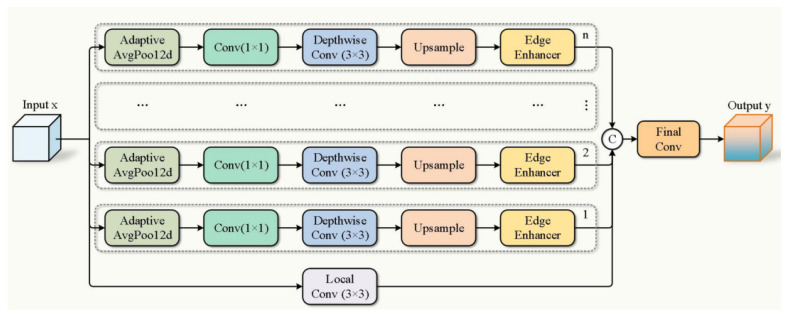
MS-EdgeNet module structure diagram.

**Figure 4 sensors-25-07049-f004:**
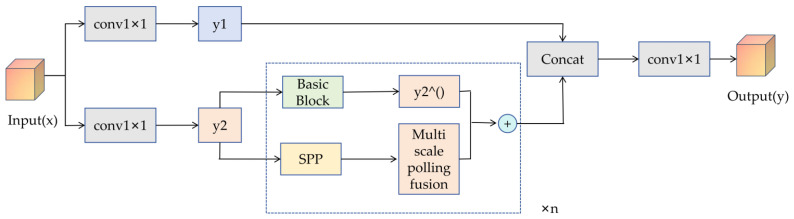
CSPStage Structure Diagram.

**Figure 5 sensors-25-07049-f005:**
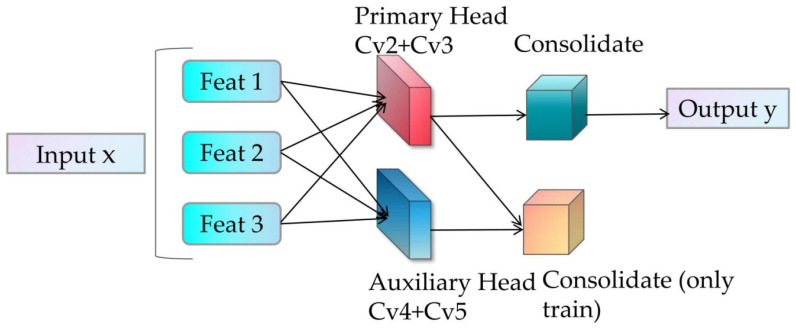
Detect-Aux Diagram.

**Figure 6 sensors-25-07049-f006:**
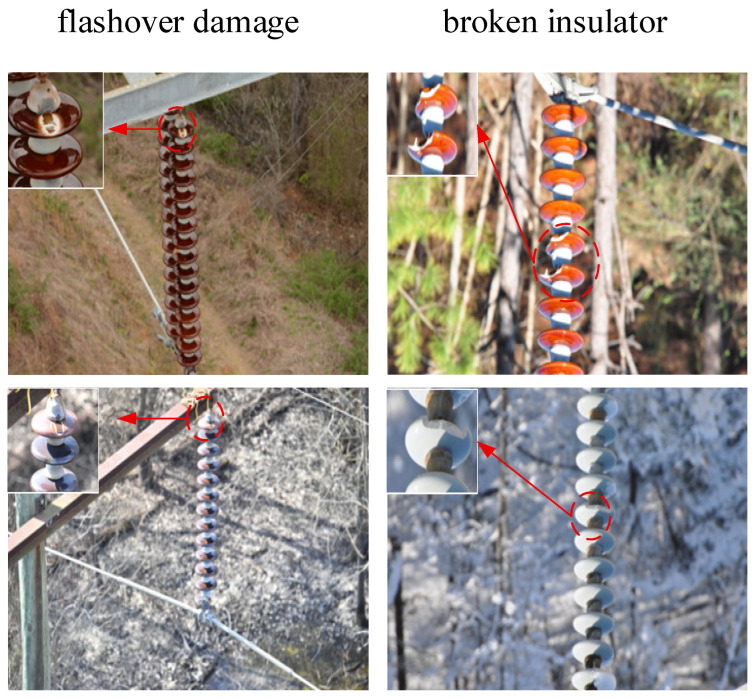
Insulator defects.

**Figure 7 sensors-25-07049-f007:**
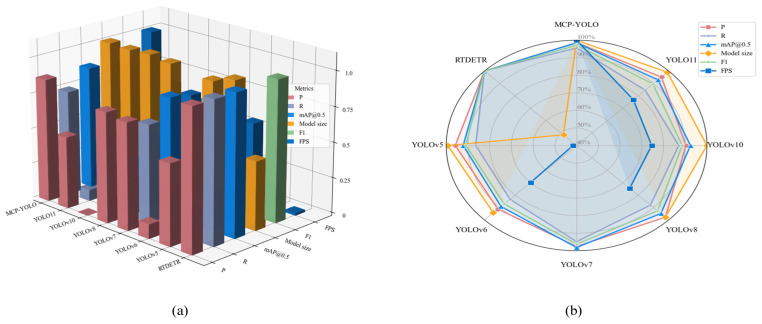
(**a**) Normalized Mainstream Algorithm Data Comparison 3d Plot (Back Normalized to Model size); (**b**) Normalized Mainstream Algorithm Data Comparison Radar Plot (Back Normalized to Model size).

**Figure 8 sensors-25-07049-f008:**
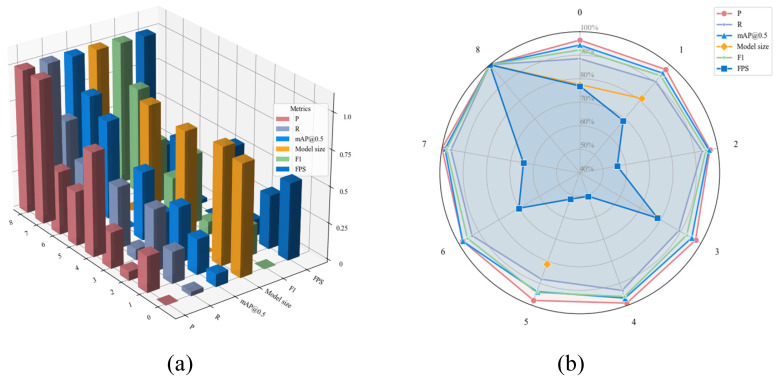
(**a**) Ablation study data comparison 3D plot (inverse normalization of Model size); (**b**) Experimental data comparison radar chart (inverse normalization of Model size).

**Figure 9 sensors-25-07049-f009:**
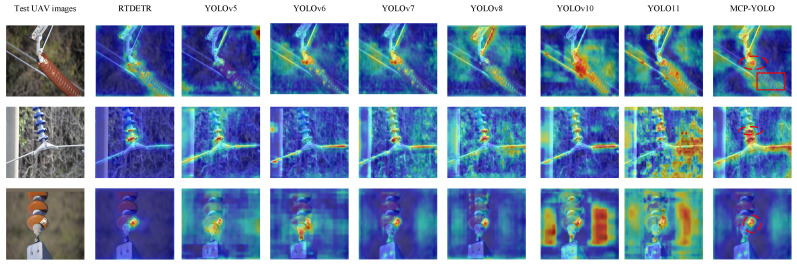
Comparison of Heatmaps Across Different Models. (Red dashed lines mark defect regions, Red solid lines mark uncontaminated regions, Erroneous red dots represent spurious background activations of baseline models.).

**Table 1 sensors-25-07049-t001:** Dataset partition details.

Dataset Split	Number of Images	Percentage
Training Set	2196	70%
Validation Set	595	20%
Test Set	300	10%
Total	3091	100%

**Table 2 sensors-25-07049-t002:** The experimental setup.

Name	System Configuration
CPU	12th Gen Intel(R) Core(TM) i7-12700KF 3.60 GHz
GPU	NVIDIA GeForce RTX 4060
Memory	16 GB
Operating system	Windows 11
Deep learning framework	Pytorch
IDE	Anaconda3
Data processing	Python3.8

**Table 3 sensors-25-07049-t003:** Comparative experiments.

Algorithm	Precision	Recall	mAP@0.5	Model Size (M)	F1score (%)	FPS
YOLOv5	0.879	0.805	0.862	9.55	84.04	104.16
YOLOv6	0.837	0.776	0.834	16.15	80.53	175
YOLOv7	0.895	0.883	0.918	139.2	88.90	80.64
YOLOv8	0.897	0.82	0.887	11.46	85.68	186.41
YOLOv10	0.828	0.81	0.864	8.64	81.89	186.41
YOLO11	0.873	0.788	0.869	9.85	82.83	192.30
RTDETR	0.917	0.931	0.935	75.81	92.39	83.22
MCP-YOLO	0.905	0.89	0.921	8.65	89.74	250

**Table 4 sensors-25-07049-t004:** Ablation Study.

Experiement Number	Base Line	C3k2-MS	DyFPN	Detect- Aux	Group SLIM	Precision	Recall	mAP @0.5	Model Size (M)	F1 Score (%)	FPS
0	√					0.873	0.788	0.869	9.85	82.83	192.30
1	√	√				0.881	0.808	0.878	9.65	84.29	172.41
2	√		√			0.875	0.83	0.886	13.99	85.19	140.84
3	√			√		0.881	0.793	0.879	9.85	83.46	196.07
4	√	√	√			0.896	0.83	0.891	13.79	86.17	126.58
5	√	√		√		0.885	0.785	0.864	9.65	84.55	129.87
6	√		√	√		0.887	0.831	0.903	13.99	85.86	175.43
7	√	√	√	√		0.905	0.855	0.909	13.79	87.93	161.29
8	√	√	√	√	√	0.905	0.89	0.921	8.65	89.74	250

**Table 5 sensors-25-07049-t005:** Experimental results of introducing the SE Block.

Experiement Number	Base Line	C3k2-MS	DyFPN	Detect- Aux	SE Block	Precision	Recall	mAP @0.5
8	√				√	0.871	0.775	0.85
9	√	√		√	√	0.868	0.818	0.891
10	√	√	√	√	√	0.887	0.864	0.903

## Data Availability

The data are not publicly available due to privacy or ethical restrictions.

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
