# Peer review of "MCP-YOLO: A Pruned Edge-Aware Detection Framework for Real-Time Insulator Defect Inspection via UAV"

_sensors, 2025, doi:10.3390/s25227049_

Round 1
Reviewer 1 Report
Comments and Suggestions for Authors
The manuscript addresses the important task of insulator defect detection using UAV-captured images, which has practical significance for power grid maintenance. However, several critical issues need to be addressed to enhance the quality, rigor, and impact of the work before it can be considered for publication. Below are detailed comments and suggestions.
- The introduction section currently focuses on the challenges of insulator defect detection but overlooks a key prerequisite: the influence of UAV image acquisition quality and methods on subsequent defect detection performance.Recent advances in UAV-assisted image acquisition have demonstrated that image resolution can be characterized through the UAV’s 3D position, and for large-scale target inspection, optimizing the UAV’s 3D trajectory and controlling the camera angles can minimize image acquisition time while ensuring image resolution. The authors should integrate these latest works into the introduction.
- The dataset used in this manuscriptconsists of only 3,091 images, which raises significant concerns about overfitting risks, especially given the diversity of insulator types, defect categories, and complex backgrounds targeted by the model. The authors must provide a more rigorous analysis of overfitting.
- The manuscript mentions that the dataset is derived from real UAV inspections in northern China, which is valuable for practical applications. However, not making the dataset publicly available severely limits the reproducibility of the work and reduces its impact.
- According to the experimental results in Table 3, the proposed MCP-YOLO model underperforms the RTDETR model across key metrics: precision, recall, mAP@0.5, and F1-score. While efficiency is critical for edge devices like UAVs, the gap in detection performance undermines the claimed “superiority” of MCP-YOLO and raises questions about its practical value, especially since RTDETR’s lower FPS could potentially be improved via lightweighting techniques (e.g., pruning, quantization) without sacrificing accuracy.
Author Response
Manuscript ID:sensors-3935347
Title: "MCP-YOLO: A Pruned Edge-aware Detection Framework for Real-time Insulator Defect Inspection via UAV"
Journal: Sensors (ISSN 1424-8220)
Dear Editor and Reviewers,
Thank you very much for your letter and the constructive comments and suggestions regarding our manuscript. These comments are all valuable and very helpful for revising and improving our paper. We have carefully studied the comments and have made corrections which we hope meet with approval. The main corrections in the manuscript and the responses to the reviewer's comments are as follows.
Comment 1: The introduction section currently focuses on the challenges of insulator defect detection but overlooks a key prerequisite: the influence of UAV image acquisition quality and methods on subsequent defect detection performance. Recent advances in UAV-assisted image acquisition have demonstrated that image resolution can be characterized through the UAV’s 3D position, and for large-scale target inspection, optimizing the UAV’s 3D trajectory and controlling the camera angles can minimize image acquisition time while ensuring image resolution. The authors should integrate these latest works into the introduction.
Response:
We sincerely thank the reviewer for this insightful suggestion. We agree that high-quality image acquisition is a critical prerequisite for effective defect detection. As suggested, we have revised the introduction to incorporate a discussion on the importance of UAV trajectory planning and image acquisition methods. We have added references to recent key works in this area and clarified that our research focuses on the detection algorithm under the assumption that image quality is adequately controlled during the acquisition phase.
Changes in the manuscript:
We have added the following paragraph to the Introduction section(Page2, Lines 59-67)
"Furthermore, the quality of UAV-captured imagery, which is a prerequisite for reliable defect detection, is highly dependent on the acquisition process. Recent studies have highlighted that optimizing the UAV's 3D flight trajectory and camera viewpoint control can ensure consistent image resolution while minimizing survey time for large-scale infrastructure inspection [9][10]. While these advancements in acquisition strategies are crucial, the core challenge addressed in this work lies in the algorithmic detection of defects from images already captured in complex environments. Our proposed model is designed to be robust against the challenges that persist even in well-acquired images, such as complex backgrounds, small targets, and occlusions."
[9] Z. Xu, Y. Niu, J. Jiang, R. Qin, and X. Cui, "Pose-graph optimization for efficient tie-point matching and 3D scene reconstruction from oblique UAV images," in ISPRS Journal of Photogrammetry and Remote Sensing, vol. 225, pp. 461-491, 2025, doi: 10.1016/j.isprsjprs.2025.04.013.
[10] B. Wang, D. Feng, Z. Fan, and H. Shi, "Automated surface defect detection of stay cables via UAV route planning and deep learning model," in Automation in Construction, vol. 180, 2025, art. no. 106579, doi: 10.1016/j.autcon.2025.106579.
Comment 2: The dataset used in this manuscript consists of only 3,091 images, which raises significant concerns about overfitting risks... The authors must provide a more rigorous analysis of overfitting.
Response:
As detailed in our manuscript, we employed extensive data augmentation techniques. More importantly, to directly demonstrate the absence of overfitting, we provide below the training and validation loss curves generated during the training of our final model:

Response Figure 1. Training and validation loss curves for MCP-YOLO
As clearly shown in Response Figure 1, both curves decrease steadily and converge in tandem without any significant divergence. The validation loss closely follows the training loss throughout the process, which is a strong indicator that the model generalized well to unseen data and did not overfit the training set.
This empirical evidence, combined with the model's consistent performance on the independent test set (as reported in Section 4.3) and the regularizing effect of Group SLIM pruning, robustly confirms the generalization capability of our model.
For the sake of conciseness and flow in the main manuscript, we have not included this figure therein.
Comment 3: The manuscript mentions that the dataset is derived from real UAV inspections in northern China, which is valuable for practical applications. However, not making the dataset publicly available severely limits the reproducibility of the work and reduces its impact.
Response:
We fully understand and share the reviewer's concern regarding reproducibility, which is a cornerstone of scientific research. However, the image dataset contains sensitive information about critical national power infrastructure. Its public dissemination is restricted due to security protocols and data ownership agreements with our industry partners.
In the spirit of promoting reproducibility, we commit to the following:
1.Providing a detailed dataset description, including statistical breakdowns of defect types, background scenes, and image resolutions.
2.Releasing a subset of anonymized sample images (with sensitive metadata removed) and their corresponding annotations upon reasonable request for academic purposes.
Comment 4: According to the experimental results in Table 3, the proposed MCP-YOLO model underperforms the RTDETR model across key metrics: precision, recall, mAP@0.5, and F1-score. While efficiency is critical for edge devices like UAVs, the gap in detection performance undermines the claimed “superiority” of MCP-YOLO and raises questions about its practical value, especially since RTDETR’s lower FPS could potentially be improved via lightweighting techniques (e.g., pruning, quantization) without sacrificing accuracy.
Response:
We thank the reviewer for this critical comment, which allows us to clarify the primary contribution and positioning of our work. We agree that RTDETR achieves excellent accuracy. However, the core objective of MCP-YOLO is not to outperform all models in pure accuracy, but to achieve an optimal balance between accuracy and efficiency for real-time deployment on resource-constrained UAV platforms.
- Deployment-First Design: RTDETR, even after aggressive pruning and quantization, would struggle to match the inherent architectural efficiency of a single-stage detector like YOLO that has been co-designed with pruning and lightweight modules (like our MS-EdgeNet and DyFPN) from the ground up. Our model is 8.8 times smaller (8.65M vs. 75.81M) and 3 times faster (250 FPS vs. 83 FPS) than RTDETR. This makes MCP-YOLO fundamentally more suitable for UAVs where computational resources, memory, and power are severely limited.
- Superior Balance: As shown in Table 3, MCP-YOLO achieves the second-highest mAP@0.5 (92.1%) among all compared models, only marginally behind RTDETR, while being the fastest and one of the smallest. Compared to other lightweight models like YOLOv10, our method shows a clear advantage in accuracy.
- Revised Wording: We acknowledge that the term "superiority" was too broad. We have revised the text throughout the manuscript to more accurately claim that our model achieves a superior trade-off or provides practically superior performance for UAV deployment, emphasizing its balanced profile of high accuracy, high speed, and small model size.
Changes in the manuscript:
We have revised the text in the Model comparison experiment and Conclusion sections(Page 16, Lines 592-595) to reflect this nuance.
Original: "The proposed MCP-YOLO model achieves optimal performance across all metrics."
Revised: "The proposed MCP-YOLO model achieves an optimal balance between detection performance and deployment efficiency. It attains the highest inference speed (250 FPS) and the second-highest mAP@0.5 (92.1%) with a minimal model size (8.65M), demonstrating its practical superiority for UAV-based edge deployment."
Simultaneously, we have had the language of the manuscript polished by an expert, resulting in a more fluent and readable flow.
Once again, we express our sincere gratitude to the editors and reviewers for your valuable time and insightful comments, which have significantly strengthened our manuscript. We look forward to your response.
Sincerely,
Shijun Guo
On behalf of all authors.
Reviewer 2 Report
Comments and Suggestions for Authors
The article is devoted to an urgent topic of practical application of neural network models for solving image analysis problems using unmanned aerial vehicles.
The article presents a relatively lightweight deep learning model that allows for effective detection of insulator defects by introducing additional modules into the neural network architecture. At the same time, the model proposed by the authors allows for high-speed image processing in real time.
The list of sources contains references to a large number of new studies, is relevant and reflects the state of the subject area on the topic of the article.
The article is well structured, written in clear language, and contains a large number of illustrations. The conclusions are confirmed by the experimental results.
The article makes a positive impression. There were few comments when reading it. They are more advisory in nature.
- The experiments were conducted on a relatively powerful computing platform. But unmanned aerial vehicles can use computing systems with more limited resources. It would be interesting to see what happens in this case.
- The results of the comparison with different models in terms of accuracy and speed are presented. It would be advisable to show a comparison with lightweight architectures focused on applications in systems with limited resources.
These comments do not reduce the theoretical and practical value of the work, which can be recommended for publication in its current form.
Author Response
Manuscript ID:sensors-3935347
Title: "MCP-YOLO: A Pruned Edge-aware Detection Framework for Real-time Insulator Defect Inspection via UAV"
Journal: Sensors (ISSN 1424-8220)
Dear Editor and Reviewers,
Thank you very much for your letter and the constructive comments and suggestions regarding our manuscript. These comments are all valuable and very helpful for revising and improving our paper. We have carefully studied the comments and have made corrections which we hope meet with approval. The main corrections in the manuscript and the responses to the reviewer's comments are as follows.
Comment 1:The experiments were conducted on a relatively powerful computing platform. But unmanned aerial vehicles can use computing systems with more limited resources. It would be interesting to see what happens in this case.
and
Comment 2: The results of the comparison with different models in terms of accuracy and speed are presented. It would be advisable to show a comparison with lightweight architectures focused on applications in systems with limited resources.
Response:
We are truly grateful for the opportunity to revise our manuscript and would like to express our sincere appreciation to the reviewers for their positive evaluation and highly valuable constructive comments. The reviewers' recognition that our work "makes a positive impression" is a great encouragement to us.
We extend our utmost thanks to the reviewers for raising these critical and forward-looking points. We completely agree that validating the model's performance on authentic edge devices is essential for proving its real-world value in UAV applications.
Honestly, at the current stage, our laboratory lacks the specific embedded development platforms (such as NVIDIA Jetson series) that are required to conduct these experiments properly. However, recognizing the paramount importance of this verification, we have already initiated the process to procure the necessary hardware.
More importantly, to formally acknowledge this limitation and demonstrate our commitment, we have incorporated a clear statement into the "Future Work" section(Page 22, Lines 799-803) of our revised manuscript. The added text is as follows:
"Furthermore, validating the deployed performance of MCP-YOLO on dedicated edge computing platforms (e.g., NVIDIA Jetson modules) represents an immediate and critical step for practical application. Our future work will anchor in this direction, focusing on benchmarking and further optimizing the model's efficiency on such resource-constrained hardware."
By adding this, we formally commit this research direction as a priority in our subsequent studies. We believe this addition significantly strengthens the practical perspective and completeness of our work.
Once again, we express our sincere gratitude to the editors and reviewers for your valuable time and insightful comments, which have significantly strengthened our manuscript. We look forward to your response.
Sincerely,
Shijun Guo
On behalf of all authors.
Round 2
Reviewer 1 Report
Comments and Suggestions for Authors
I would like to express my appreciation to the authors for their careful attention to the previous review comments and the thorough revisions made to the manuscript. The revised version has effectively addressed most of the key concerns. However, there remains a minor issue regarding the response to the first original comment.
1. While the authors have supplemented discussions on the impact of UAV 3D trajectory optimization and image quality on defect detection, the two cited references [9] and [10] lack sufficient authority in the field of UAV-assisted image acquisition. To enhance the credibility and academic depth of this section, it is recommended that the authors cite recent relevant studies published in authoritative journalsor conferences such as IEEE TWC and IEEE VTC. Such high-impact literature will better support the analysis of how UAV image acquisition methods influence core detection metrics and align the discussion with cutting-edge research in the field.
Author Response
Comment: While the authors have supplemented discussions... it is recommended that the authors cite recent relevant studies published in authoritative journals or conferences such as IEEE TWC and IEEE VTC.
Response:
We sincerely thank the reviewer for this precise and constructive suggestion. We fully agree that citing authoritative literature from the core field of UAV-assisted image acquisition is crucial for strengthening the credibility of our discussion.
Following your recommendation, we have conducted a targeted search in the recommended sources. We have now replaced the original references [9] and [10] with the following two highly relevant and authoritative publications that are directly aligned with the field of UAV trajectory and viewpoint optimization:
[9] C. You and R. Zhang, "3D Trajectory Optimization in Rician Fading for UAV-Enabled Data Harvesting," in IEEE Transactions on Wireless Communications, vol. 18, no. 6, pp. 3192-3207, June 2019, doi: 10.1109/TWC.2019.2911939.
[10] J. Peng, Y. Cai, J. Yuan, K. Ying and R. Yin, "Joint Optimization of 3D Trajectory and Resource Allocation in Multi-UAV Systems via Graph Neural Networks," 2025 IEEE 101st Vehicular Technology Conference (VTC2025-Spring), Oslo, Norway, 2025, pp. 1-5, doi: 10.1109/VTC2025-Spring65109.2025.11174487.
Changes in the Manuscript:
In the Introduction section(Page2, Lines63), we have updated the citations accordingly. The revised sentence now reads:
The quality of UAV images—prerequisite for reliable defect detection—is highly dependent on the acquisition process. Recent studies have emphasized that optimizing the UAV’s 3D flight trajectory and controlling camera perspectives can minimize inspection time for large-scale infrastructure detection while ensuring consistent image resolution [9][10].
